# Cost-effectiveness of internet-delivered cognitive behavioural therapy in patients with cardiovascular disease and depressive symptoms: secondary analysis of an RCT

Ghassan Mourad  ,[1] Johan Lundgren,[1] Gerhard Andersson,[2,3,4] Magnus Husberg,[1] Peter Johansson[1,5]

¹Department of Health, Medicine and Caring Sciences, Linköping University, Linköping, Sweden
²Department of Behavioural Sciences and Learning, Linköping University, Linköping, Sweden
³Department of Biomedical and Clinical Sciences, Linköping University, Linköping, Sweden
⁴Department of Clinical Neuroscience, Karolinska Institutet, Stockholm, Sweden
⁵Department of Internal Medicine, Linköping University, Linköping, Sweden

**Correspondence to**
Dr Ghassan Mourad;
ghassan.mourad@liu.se

## ABSTRACT

**Introduction** Cost-effectiveness evaluations of psychological interventions, such as internet-delivered cognitive behavioural therapy (iCBT) programmes, in patients with cardiovascular disease (CVD) are rare. We recently reported moderate to large effect sizes on depressive symptoms in CVD outpatients following a 9-week iCBT programme compared with an online discussion forum (ODF), in favour of iCBT. In this paper, we evaluate the cost-effectiveness of this intervention.

**Methods** Cost-effectiveness analysis of a randomised controlled trial. The EQ-5D-3L was used to calculate quality-adjusted life-years (QALYs). Data on healthcare costs were retrieved from healthcare registries.

**Results** At 12-month follow-up, the QALY was significantly higher in iCBT compared with the ODF group (0.713 vs 0.598, p=0.007). The mean difference of 0.115 corresponds with 42 extra days in best imaginable health status in favour of the iCBT group over the course of 1 year. Incremental cost-effectiveness ratio (ICER) for iCBT versus ODF was €18 865 per QALY saved. The cost-effectiveness plane indicated that iCBT is a cheaper and more effective intervention in 24.5% of the cases, and in 75% a costlier and more effective intervention than ODF. Only in about 0.5% of the cases, there was an indication of a costlier, but less effective intervention compared with ODF.

**Conclusions** The ICER of €18 865 was lower than the cost-effectiveness threshold range of €23 400–€35 100 as proposed by the NICE guidelines, suggesting that the iCBT treatment of depressive symptoms in patients with CVD is cost-effective.

**Trial registration number** NCT02778074; Post-results.

### Strengths and limitations of this study

► This is the first cost-effectiveness study of internet-delivered cognitive behavioural therapy (iCBT) in patients with cardiovascular disease and depression.
► The study was based on actual costs retrieved from data registries.
► The study was primarily designed to evaluate the effect of iCBT and not cost-effectiveness.

including patients with chronic stable angina, depression was a significant predictor of increased healthcare costs 1 year after index angiogram. The adjusted cumulative healthcare cost was 1.33, indicating a mean of 33% increase in 1 year healthcare costs for those with depression. In another study, Palacios *et al*[4] measured depression over a 3-year period in outpatients with coronary heart disease and found that patients with worsening or chronic depressive symptoms had more than twice the costs than patients without depressive symptoms. However, they also found that those with depressive symptoms who were detected and treated had lower costs than those who remained untreated. The authors emphasised that interventions targeting depression and healthcare costs in outpatients are warranted.[4] In addition to this, the cost-effectiveness of psychological interventions in cardiac rehabilitation (CR) programmes has been reported to be the least studied CR component.[5] In conclusion, this highlights a need for studies that perform health-economic/cost-effectiveness evaluations of psychological interventions in patients with CVD.

In recent years, internet-delivered cognitive behavioural therapy (iCBT) has received increased attention as a treatment option for

## INTRODUCTION

Depression is found in 20%–40% of patients with cardiovascular disease (CVD),[1] and has shown to shorten life expectancy,[1] induce higher risk of non-fatal cardiac events and re-hospitalisation, compared with patients with CVD without depression.[2] Depression in CVD is also a societal problem in the way that it increases the healthcare costs. In a study[3]

depression in patients with CVD.[6 7] One reason for this is a low access to face-to-face CBT. Another reason is the COVID-19 pandemic which has increased the need to deliver care by digital solutions, in particular for persons who are susceptible of a worser outcome in COVID-19, such as those with CVD.[6] In general, for middle-aged populations iCBT has been found to decrease depression,[8] but only a minority of all iCBT studies have performed cost-effectiveness analyses. Moreover, in the existing studies the evidence of the cost-effectiveness of iCBT on depression is mixed.[9–11] In CVD, only a few studies have evaluated the effect of iCBT on depression[4] and to our knowledge no study has evaluated the cost-effectiveness of iCBT in patients with CVD. In a recent randomised controlled trial (RCT), we found that a 9-week iCBT-programme had moderate to large effect on depressive symptoms in CVD outpatient compared with an online discussion forum (ODF), where new discussion topics were presented each week over a 9-week period.[9] The ODF was chosen as a comparator as it is recommended to use active controls in iCBT studies.[12] The aim of this study was to report the cost-effectiveness of iCBT compared with ODF.

## METHODS

### Design, setting and participants

This was a cost-effectiveness study of an RCT aiming to reduce depressive symptoms in patients with CVD, and that has been described in detail elsewhere.[9] In brief, patients who had been in contact with the medical or cardiac clinics at four hospitals in Southeastern Sweden (N=11 992) were invited to participate by post. A total of 272 registered their interest and were screened for depressive symptoms (ie, score ≥5 on the Patient Health Questionnaire-9). In total 144 patients fulfilled the eligibility criteria of individuals age ≥18 years and had no hospitalisations during the past 4 weeks prior to inclusion. Patients were randomly allocated to 9 weeks of iCBT (n=72) or ODF (n=72). Thirty-eight per cent (n=27) of those in the ODF group received iCBT after study completion. The iCBT programme was guided by study nurses and comprised of goal setting, psychoeducation, problem-solving, behavioural activation including homework assignments with weekly written feedback. In the iCBT group, 60% of the participants completed all seven modules, and 82% completed at least four. In the ODF, nine discussion topics were included and moderated by the study nurses. About 27% of those participants were active at least nine times during the intervention.

### Health-related quality of life and quality-adjusted life-years

HRQoL was measured using the EQ-5D-3L (three levels), which consists of a descriptive system and the EQ Visual Analogue Scale (EQ-VAS). The EQ-5D-3L descriptive system comprises five dimensions: mobility, self-care, usual activities, pain/discomfort and anxiety/depression. Each dimension has three levels: no problems, some problems and extreme problems with scores between 1 and 3 for each dimension. The EQ-VAS displays the patient's self-rated health on a vertical visual analogue scale with scores between 0 (worst imaginable health state) and 100 (best imaginable health state). The scores from the dimensions are used to calculate an index value that reflects how good or bad a health state is in relation to the general population of a country, and this facilitates the calculation of quality-adjusted life-years (QALYs) that are used to inform economic evaluations of healthcare. The UK value set was used for the calculation of the EQ-5D-3L index.[13]

### Costs

The healthcare perspective was used for the cost analyses. Cost data were retrieved using care data registries in the three regions in Sweden where the study was performed, that is, Östergötland, Jönköping and Kalmar. The retrieved data comprised all healthcare use and costs for outpatient clinic/primary care contacts and hospital admissions, and most private practices. The information is based on data from administrative registries where all data regarding healthcare use are stored. Six participants included in the primary RCT were residents in other regions that precluded costs data to be obtained. Hence, these patients are excluded from the analysis in this study.

### Cost-effectiveness

Cost-effectiveness was assessed with an incremental cost-effectiveness ratio (ICER). The costs and effects included in the ICER are from baseline to 1-year postintervention, but also costs for guidance and support to patients in iCBT and ODF groups. The ICER can be seen as the additional costs needed to gain an additional QALY by the iCBT compared with ODF.

### Patient and public involvement

It was not possible to involve patients or the public in the design, conduct, and reporting of this study as this was mainly based on registry data.

Demographic data and healthcare costs were presented using numbers, percentages, mean values, and SD. $\chi^2$ test was used for categorical variables and Student's t-test for continuous variables when analysing differences in demographic data between the iCBT and ODF groups.

To calculate the QALYs, missing data regarding HRQoL index value was imputed using last observation carried forward. The average index value was then multiplied by the time between each measurement (ie, baseline to 9 weeks, 9 weeks to 6 months and 6 months to 12 months) and the 1 year QALY was the sum of the four QALY values. Student's t-test was then used to compare the QALY between the groups. When comparing the difference in EQ-VAS over 12 months between the groups, analysis of covariance was used which allows adjusting for baseline scores and regression to the mean.

Costs were in Swedish Kronor and were converted into Euros using the rate of May 2021 (ie, 1 Euro=10.50 Swedish Kronor). Costs for the iCBT and ODF were

**Table 1**  Demographic data of patients randomised to Internet-delivered cognitive behavioural therapy (iCBT) or online discussion forum (ODF) or excluded due to no cost data

|  | iCBT (n=70) | ODF (n=68) | Excluded (n=6) |
|---|---|---|---|
| Sex, n (%) |  |  |  |
| Male | 46 (66) | 40 (59) | 3 (50) |
| Female | 24 (34) | 28 (41) | 3 (50) |
| Age, mean (SD) | 62 (12) | 64 (12) | 60 (19) |
| Educational level, n (%) |  |  |  |
| Elementary | 7 (10) | 9 (13) | 3 (50) |
| Upper secondary/high school | 15 (21) | 21 (31) | 1 (17) |
| Postsecondary vocational education | 12 (17) | 6 (9) | 0 (0) |
| College/university | 36 (51) | 32 (47) | 2 (33) |
| Work status, n (%) |  |  |  |
| Working | 25 (36) | 17 (25) | 2 (33) |
| Sick leave/disability pension | 7 (10) | 10 (15) | 1 (17) |
| Retired | 32 (46) | 33 (49) | 3 (50) |
| Other (including unemployed and students) | 6 (9) | 8 (12) | 0 (0) |
| Marital status, n (%) |  |  |  |
| Married/cohabiting | 52 (74) | 50 (74) | 4 (67) |
| Living alone | 18 (26) | 18 (26) | 2 (33) |
| Financial situation, n (%) |  |  |  |
| Very good | 10 (14) | 10 (15) | 1 (17) |
| Good | 48 (69) | 43 (63) | 3 (50) |
| Problematic | 9 (13) | 13 (19) | 2 (33) |
| Very problematic | 3 (4) | 2 (3) | 0 (0) |
| Smoking, n (%) |  |  |  |
| Never | 32 (46) | 34 (50) | 3 (50) |
| Ex-smoker | 36 (51) | 31 (46) | 3 (50) |
| Smoker | 2 (3) | 3 (4) | 0 (0) |
| Alcohol, n (%) |  |  |  |
| 0–4 units per week | 49 (70) | 54 (79) | 6 (100) |
| 5–9 units per week | 17 (24) | 10 (15) | 0 (0) |
| 10–14 units per week | 3 (4) | 4 (6) | 0 (0) |
| 15 or more units per week | 1 (1) | 0 (0) | 0 (0) |
| Depressive symptoms and HRQoL |  |  |  |
| PHQ-9, mean (SD) | 10.6 (4.5) | 10.1 (5.0) | 12.9 (5.0) |
| EQ-VAS mean (SD) | 52.8 (20.0) | 56.2 (18.0) | 72.5 (13.0)* |
| Type of CVD |  |  |  |
| Ischaemic heart disease, n (%) | 34 (49) | 28 (41) | 1 (17) |
| Atrial fibrillation, n (%) | 38 (54) | 39 (57) | 4 (67) |
| Heart failure, n (%) | 18 (26) | 18 (26) | 2 (33) |
| Self-reported primary heart-related health problem |  |  |  |
| Ischaemic heart disease, n (%) | 26 (37) | 23 (34) | 0 |
| Atrial fibrillation, n (%) | 32 (46) | 29 (43) | 4 (67) |
| Heart failure, n (%) | 12 (17) | 16 (24) | 2 (33) |
| Co-morbidities (three or more), n (%) | 10 (14) | 13 (19) | 0 (0) |

*Significantly higher HRQoL at baseline in those excluded due to no healthcare cost data.
CVD, cardiovascular disease; EQ-VAS, EQ Visual Analogue Scale; HRQoL, health-related quality of life; PHQ-9, Patient Health Questionnaire-9.

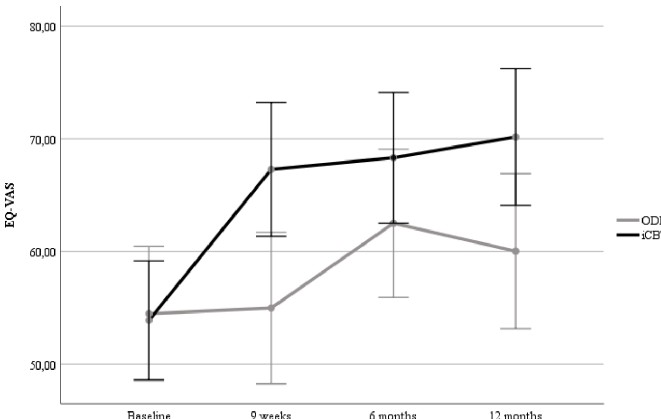

**Figure 1** EQ-VAS measured before and 9 weeks, 6 and 12 months after the intervention in the Internet-delivered cognitive behavioural therapy (iCBT) and online discussion forum (ODF) groups. EQ-VAS, EQ Visual Analogue Scale.

estimated based on the mean salary of a nurse (ie, €5440) including cost surcharges, resulting in a cost per hour of €34. The time needed to guide and support patients were in total 140 hours for the iCBT group and 11 hours for the ODF group, leading to total costs of €4760 and €374, respectively. The costs for outpatient clinic/primary care contacts and hospital admissions are reported as costs for materials (including drugs and materials and medical devices), staff (including staff and premises), medical services (including radiology and laboratory services) and surgery (including anaesthesia, surgery, postoperative care and intensive care). In 205 (2.3%) of a total of 8893 outpatient clinic/primary care contacts, the costs were not specified. These costs are therefore reported under staff costs since most of the costs for the outpatient/primary care contacts were related to staff costs. Cost data concern the years 2017–2019 regardless of inflation.

Cost-effectiveness as assessed by the ICER was calculated by dividing the incremental costs by the incremental effects between the groups. The ICER was bootstrapped five thousand times to plot the cost-effectiveness plane and cost-effectiveness acceptability curve to reveal the uncertainty around the results.[14] These were made in Microsoft Excel version 2201.

The IBM SPSS V.25.0 was used for data analysis. The level of p<0.05 was used to determine statistically significant differences.

## RESULTS

### Study participants

The sample in the original RCT consisted of 144 participants. Data on healthcare cost were available for 138 participants (n=70 in the iCBT group and n=68 in the ODF group), 86 men and 52 women with a mean age of 63 years (SD 12.0) (table 1). There were no significant differences between the iCBT and ODF groups in demographic data or baseline levels of depression and HRQoL. The six participants with no healthcare cost data had significantly better baseline HRQoL (p=0.023) compared with the others, but no other statistically significant differences were found in the other variables in table 1.

### Health-related quality of life and quality of adjusted years

Data on HRQoL was collected from 69 patients in the iCBT group and 67 patients in the ODF group at baseline, and from 58 and 50 patients in each group at 12-month follow-up. The iCBT group reported an improvement in EQ-VAS from 53.9 at baseline to 70.2 at 12 month follow-up, while the ODF group improved from 54.5 to 60.0 during the same period (figure 1). The iCBT group continued to improve, although slightly, over time, while the ODF group reported their best health status at 6-month follow-up and decreased at 12 months. The difference in EQ-VAS at 12-month follow-up was 10.2 points in favour of the iCBT group, which was statistically significant (p=0.006).

The QALY values differed significantly (p=0.007) between the iCBT and the ODF groups who had 0.713 and 0.598 QALYs, respectively. The mean difference of 0.115 corresponds with 42 days in best imaginable health status in favour of the iCBT group over the course of 1 year.

### Costs

The costs for hospital admissions and outpatient clinic/primary care contacts for the iCBT and the ODF groups

**Table 2** Healthcare costs in Euro per patient randomised to Internet-delivered cognitive behavioural therapy (iCBT) or online discussion forum (ODF) 1 year before, during (9 weeks) and 1-year postintervention

| | Hospital admission costs | | Out-patient/primary care costs | | Total costs | | |
|---|---|---|---|---|---|---|---|
| | iCBT (n=70) | ODF (n=68) | iCBT (n=70) | ODF (n=68) | iCBT (n=70) | ODF (n=68) | P value |
| One year before intervention m (SD) | 3964 (8193) | 5255 (9159) | 3261 (2911) | 3946 (3467) | 7225 (9639) | 9201 (10639) | 0.255 |
| 9 weeks during intervention m (SD) | 445 (1431) | 491 (1874) | 504 (657) | 620 (779) | 949 (1755) | 1111 (2242) | 0.636 |
| One-year postintervention m (SD) | 4472 (22030) | 1886 (5281) | 3054 (3111) | 3365 (3348) | 7526 (22728) | 5250 (6792) | 0.430 |

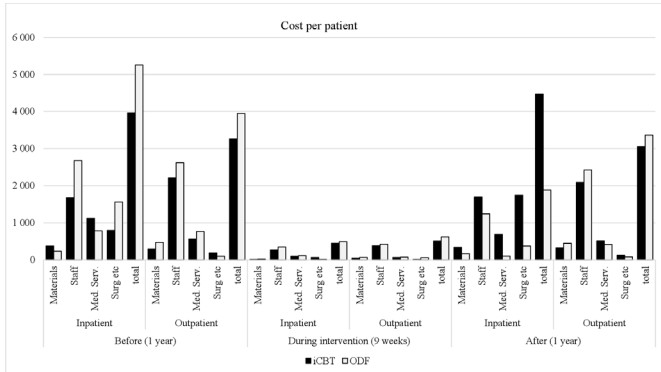

**Figure 2** Mean cost per patient randomised to Internet-delivered cognitive behavioural therapy (iCBT, n=70) or online discussion forum (ODF, n=68) 1 year before, during (9 weeks) and 1-year postintervention.

are presented in table 2. The hospital admission costs were lower in the iCBT group compared with the ODF group the year before (€3964 vs €5255) and during the intervention (€445 vs €491), but not the year after intervention (€4472 vs €1886). Similar findings were found regarding the out-patient costs the year before (€3261 vs €3946), as well as during the intervention (€504 vs €620). However, the outpatient costs were also lower in the iCBT group the year after intervention (€3054 vs €3365). None of the differences were statistically significant.

Figure 2 displays the total costs divided into material, staff, medical services, and surgery costs. No statistically significant differences were found between the groups in any of the areas.

### Cost-effectiveness

The cost-effectiveness results from a healthcare perspective are presented in table 3. The ICER for iCBT versus ODF was €18865 per QALY saved.

The incremental cost-effectiveness plane and the acceptability curve for cost per QALY are presented in figure 3. In about 24.5% of the cases the cost-effectiveness plane indicated a cheaper but more effective intervention, and in 75% a costlier and more effective intervention than ODF. Only in less than 0.5% of the cases, there was an indication of a costlier, but less effective intervention compared with ODF. The cost-effectiveness acceptability curve indicates that iCBT is cost-effective compared with ODF, with a 50% probability of the iCBT being cost-effective at a willingness to pay threshold of €15905 per QALY.

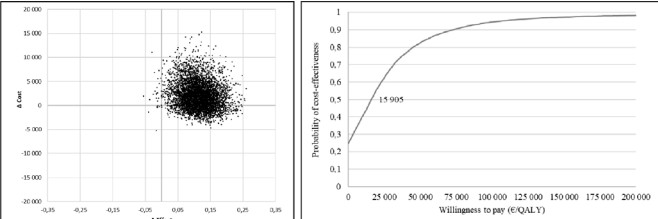

**Figure 3** Cost-effectiveness plane and acceptability curve for cost per QALY. QALY, quality-adjusted life-year.

### DISCUSSION

This study evaluated the cost-effectiveness of iCBT compared with ODF in patients with CVD and depressive symptoms. We found that iCBT led to sustained improvement in HRQoL at 12-month follow-up. Although there are no definite criteria to define what is to be considered as cost-effective in the Swedish healthcare system, the ICER of €18865 can be seen as cost-effective in comparison with the cost-effectiveness threshold range of (£20 000–£30 000, or converted to Euro approximately €23 400–€35 100) as proposed by the NICE guidelines.[15]

The cost-effectiveness of iCBT as a treatment for depression has been reported in a systematic review of 12 studies.[16] Seven of the reviewed studies reported an ICER for the intervention ranging between €3088 and €22609 and were deemed as cost-effective compared with the control condition according to the NICE threshold range.[15] At the 12-month follow-up, we found significantly better HRQoL and consequently better QALYs in the iCBT group. Our ICER of €18865 can be compared with other iCBT studies[16] and thus be seen as cost-effective. Moreover, we also found a 50% probability that iCBT compared with ODF was cost-effective at a willingness to pay threshold of €15905 per QALY. This result is similar to another iCBT study treating depression in patients with diabetes mellitus and reporting a 51% probability of cost-effectiveness to a willingness to pay of €14 000.[17]

At the 12-month follow-up, we could however not detect any significant reduction in healthcare costs of iCBT compared with ODF. Patients with CVD and depressive symptoms seem to be more reluctant to perform self-care and to seek care when needed.[18] On the other hand, improvement in depressive symptoms in patients with CVD has been found to be associated with an increase in self-care.[19] Thus, one explanation is that patients became less depressed following iCBT and by this more active in their own self-care, for example seeking healthcare when

**Table 3** Cost-effectiveness of internet-delivered cognitive behavioural therapy (iCBT) compared with online discussion forum (ODF)

|  | iCBT | ODF | Incremental difference |
|---|---|---|---|
| Cost for healthcare use (Euro) | 8541.3 | 6366.7 | 2174.6 |
| Effect (QALY) | 0.71277 | 0.59750 | 0.11527 |
| Cost-effectiveness (ICER) |  |  | 18865.3 |

ICER, incremental cost-effectiveness ratio; QALY, quality-adjusted life-year.

that was needed. Another explanation for not finding any differences between the groups in healthcare costs could be because patients in the ODF group also were offered iCBT post intervention due to ethical reasons, and 38% of them accepted and took part of the intervention. This may have impacted the possibility to detect differences between the groups at the 12-month follow-up.

In the review by Paganini et al,[16] the mean age of the populations in 11 of the 12 studies ranged between 30 and 50 years and none of the populations were included based on having any somatic disease. The patients with CVD in the current study who had a mean age of 63 years are at higher risk than younger patients without a somatic disease to suffer a deterioration in their physical health despite improvement in depressive symptoms by iCBT or not. After scrutinising the costs, we found one participant in the iCBT group who had received intensive care for an extended period to a cost of € 200 000. A psychological intervention (eg, iCBT) may not be expected to have any major effect on the pathophysiologic progress of the CVD compared with pharmacological (eg, angiotensin receptor neprilisyn inhibitor) or technical interventions (eg, pacemaker). Therefore, another explanation could be that when evaluating cost-effectiveness of psychosocial interventions in patients with chronic somatic diseases one may have to expect a limited possibility to detect reductions in healthcare costs.

CR can be seen as the cornerstone in the management of CVD with the goal to improve quality of life and to reduce morbidity and mortality in these patients.[20] This is achieved by supervision and education to teach patients to improve their self-care and to modify CVD risk factors by, for example, stop smoking, eating healthy diets, and becoming physically active. Depression is another modifiable risk factor in CVD.[21] Despite this, a recent systematic review of 19 studies studying the cost-effectiveness of CR reported that psychological interventions were the least studied CR components and they concluded that today there is limited evidence for the cost-effectiveness of such interventions in CR.[5] Moreover, to our knowledge, no studies have evaluated the cost-effectiveness of Internet-based psychological interventions aimed to treat depressive symptoms in patients with CVD. It is, therefore, not easy to compare our data with other studies in CVD. However, four studies included in the review of Shields et al[5] evaluated the cost-effectiveness of CR as a telehealth intervention. They reported ICERs for CR as telehealth between dominant and €21 200 for three of the studies, which were deemed to be cost-effective. The fourth study had an ICER of €512 198 and was therefore not cost-effective. An explanation for the huge ICER is that the study compared an intervention that was more expensive but had little incremental effect compared with the control. Our ICER of €18 865 for iCBT to treat depressive symptoms in patients with CVD is comparable to the cost-effectiveness for CR performed as telehealth.

Guided iCBT seems to be more effective than unguided iCBT.[16] Furthermore, an additional advantage when treating patients with CVD is if iCBT is guided by healthcare personnel with experience in CVD care, such as nurses, and a brief education in CBT.[9] This could certainly facilitate the implementation of iCBT in CVD care. A drawback with guidance is that policy-makers at first sight can see this to drive the costs for delivering iCBT. On the other hand, Sheilds et al[5] discussed that it is more likely that the costs will be reduced when scaling up a telehealth intervention in a clinical setting, for example that initial investments can be spread over a larger number of patients. To summarise, we have in previous studies reported that iCBT improves depressive symptoms as well as physical activity and HRQoL in patients with CVD.[9 22] This study also shows that iCBT is a cost-effective risk factor modification tool that can be used in the clinical management of depressive symptoms in patients with CVD.

## Limitations

A major limitation with this study is that it was primarily designed to evaluate the effect of iCBT on depression, and thus not designed for health economic evaluation. However, it is not uncommon that healtheconomic evaluations are performed as secondary data analysis of RCT studies. It is therefore possible that the healtheconomic benefits of iCBT have been underestimated in our study.

## CONCLUSIONS

iCBT for depression in patients with CVD can be cost-effective and implementation into CR could be feasible. However, before realisation more studies evaluating treatment and cost-effectiveness of iCBT in patients with CVD are needed.

**Contributors** GM is the guarantor for the work. GM, JL, GA and PJ contributed to the conception and design of the study. GM, JL and MH collected and processed the data, and performed statistical analyses. GM, JL, GA, MH and PJ contributed to the analysis and interpretation of the data and drafting of the manuscript.

**Funding** The study was funded by the Swedish Research Council (2015-02600) and the Medical Research of Southeast Sweden (FORSS-848511).

**Competing interests** None declared.

**Patient and public involvement** Patients and/or the public were not involved in the design, or conduct, or reporting, or dissemination plans of this research.

**Patient consent for publication** Not applicable.

**Ethics approval** The Regional Ethical Review Board in Linköping, Sweden (ref. no. 2011/166-31).

**Provenance and peer review** Not commissioned; externally peer reviewed.

**Data availability statement** Data are available on reasonable request.

**ORCID iD**
Ghassan Mourad http://orcid.org/0000-0001-9140-8922

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
