## [Reviewer comments · BMJ Open]

ARTICLE DETAILS

TITLE (PROVISIONAL)	Cost-effectiveness of Internet-delivered cognitive behavioural therapy in patients with cardiovascular disease and depressive symptoms. Secondary analysis of a RCT.
AUTHORS	Mourad, Ghassan; Lundgren, Johan; Andersson, Gerhard; Husberg, Magnus; Johansson, Peter

VERSION 1 – REVIEW

REVIEWER	Tyrer, Peter Imperial College London, Psychological medicine
REVIEW RETURNED	05-Jan-2022

GENERAL COMMENTS	The findings of this study are of value. I am particularly impressed by the inclusion of a highly expensive patient in the iCBT group that led to extra expense; many investigators might have justified the exclusion of this patient using standard statistical arguments for outliers. The study was not planned as a cost-effectiveness study and so there are some deficiencies here but not enough to be a problem with publication. The cost data come from health care registries but the reader is left in the dark about the accuracy and comprehensiveness of these data. Are all contacts in primary care included, and are private consultations? There is also no information on the proportion of patients accepted for the study. Many of these studies find it difficult to recruit. How many people were approached to take part and what proportion constituted those randomised? I am not a health economist but my impression was that these analyses were carried out appropriately.
--

REVIEWER	Natsky, Andrea Flinders University, Health Economics, College of Medicine and Public Health
REVIEW RETURNED	07-Jan-2022

GENERAL COMMENTS	This study provides important finding for economic evaluation of iCBT-I in CVD and depressive patients. I believe the study has potential to contribute to the literature and change in health policy and practices. The authors did a good job following the CHEERS guideline. I would like to recommend to the Editor to consider accepting the paper provided the following points are addressed:
--

	Major Page 6, line 12. A brief foreword about ODF is beneficial for reader to be included in the introduction. Also, an additional explanation of why ODF is used as comparator in this study can be included in the methods section. Page 8. Please provide references for methods used (t-test, QALY + ICER calculations appropriately). Page 11. Please re-structure the costs section. Separate the findings in the first sentence and provide numbers from the table. Also, please remove the explanation to discussion section (“this is due... line 45”) Can the authors explain the justification for using UK NICE threshold for the current setting? Page 15, line 17. Can the authors further elaborate on the different findings of ICER 512,198? The result regarding CEAC is shown for WTP threshold of 15,905 per QALY at 0.5 probability of cost-effectiveness. It would be helpful to show in the Figure as well as in the main text (result section, and comments on the findings on discussion section accordingly) the probability of cost-effectiveness at the current WTP threshold used (UK NICE). Minor Page 5, line 18. Please include reference of the study indicated in the sentence. Page 6, line 37. Revise the sentence to: In total, 144 patients fulfilled the eligibility criteria of individuals age>18 years and no hospitalizations during the past four weeks prior to inclusions. Participants were (randomly?) allocated to nine weeks of iCBT (n=72) or ODF (n=72). Page 6, line 37. Add of : Comprised “of” goal setting... Page 7, line 41. Consider changing the sentence to: Six participants included in the primary RCT were residents in other regions that precluded costs data to be obtained. Hence, these patients are excluded from the analysis in this study. This sentence is also supposed to be placed in the result, not the method section. Page 11 line 13, place text (Figure 1) by end of sentence. Page 14 line 20, change our to ours Page 14, line 19. Provide reference for this sentence. Throughout, please make sure to place zero before point and make it consistent. E.g., change .05 to 0.05. Throughout, please carefully proofread for other grammatical errors, alignment and placing of references/ table/ figure by end of sentences.
--	--

VERSION 1 – AUTHOR RESPONSE

Response to reviewers:

Reviewer: 1

The findings of this study are of value. I am particularly impressed by the inclusion of a highly expensive patient in the iCBT group that led to extra expense; many investigators might have justified the exclusion of this patient using standard statistical arguments for outliers.

- Response: Thank you for this comment, we appreciate your support for this choice.

The study was not planned as a cost-effectiveness study and so there are some deficiencies here but not enough to be a problem with publication. The cost dates come from health care registries but the reader is left in the dark about the accuracy and comprehensiveness of these data. Are all contacts in primary care included, and are private consultations?

- Response: Thank for this comment. We have now clarified where data was retrieved and also added an explanatory text to ensure the accuracy of our data on page 6:

The retrieved data comprised all healthcare use and costs for outpatient clinic/primary care contacts and hospital admissions, and most private practices. The information is based on data from administrative registries where all data regarding healthcare use are stored.

There is also no information on the proportion of patients accepted for the study. Many of these studies find it difficult to recruit. How many people were approached to take part and what proportion constituted those randomised?

- Response: We agree that recruiting participants to such studies is a challenge. However, we have learned that by approaching prospective participants by letters and informing about the study, has been a feasible way to recruit. We have now added the numbers that were lacking to enhance understandability. The following text is modified on page 5:

In brief, patients who had been in contact with the medical or cardiac clinics at four hospitals in Southeastern Sweden (N=11,992) were invited to participate by post. A total of 272 registered their interest and were screened for depressive symptoms (i.e., score ≥ 5 on the Patient Health Questionnaire-9 (PHQ-9)). In total 144 patients fulfilled the eligibility criteria of individuals age ≥ 18 years and had no hospitalizations during the past four weeks prior to inclusion. Patients were randomly allocated to nine weeks of iCBT (n=72) or ODF (n=72).

I am not a health economist but my impression was that these analyses were carried out appropriately.

- Response: Thank you!

Reviewer: 2

This study provides important findings for the economic evaluation of iCBT-I in CVD and depressive patients. I believe the study has the potential to contribute to the literature and change in health policy and practices. The authors did a good job following the CHEERS guideline.

I would like to recommend to the Editor to consider accepting the paper provided the following points are addressed:

- Response: Thank you for this comment.

Major

1. Page 6, line 12. A brief foreword about ODF is beneficial for the reader to be included in the introduction. Also, an additional explanation of why ODF is used as a comparator in this study can be included in the methods section.

- Response: We have now clarified this on page 5, as described below:

In a recent RCT we found that a 9-week iCBT-program had moderate to large effect on depressive symptoms in CVD outpatient compared to an online discussion forum (ODF), where new discussion topics were presented each week over a 9-week period [1]. The ODF was chosen as a comparator as

it is recommended to use active controls in iCBT studies [12]. The aim of this study was to report the cost-effectiveness of iCBT compared to ODF.

2. Page 8. Please provide references for methods used (t-test, QALY + ICER calculations appropriately).

-Response: We have inserted a reference for the ICER.

3. Page 11. Please re-structure the costs section. Separate the findings in the first sentence and provide numbers from the table. Also, please remove the explanation to the discussion section (“this is due... line 45”)

-Response: Thank you for this comment. We have now restructured and re-written the whole section and made it clearer. See below:

The costs for hospital admissions and outpatient clinic/primary care contacts for the iCBT and the ODF groups are presented in Table 2. The hospital admission costs were lower in the iCBT group compared to the ODF group the year before (€3964 vs. €5255) and during the intervention (€445 vs. €491), but not the year after intervention (€4472 vs. €1886). Similar findings were found regarding the out-patient costs the year before (€3261 vs. €3946), as well as during the intervention (€504 vs. €620). However, the out-patient costs were also lower in the iCBT group the year after intervention (€3054 vs. €3365). None of the differences were statistically significant.

4. Can the authors explain the justification for using the UK NICE threshold for the current setting?

-Response: Unfortunately, there are no such cost-effectiveness thresholds in the Swedish system. We therefore used the UK threshold to better compare and discuss our findings. This because the EQ5D UK value set is the standard to use for QALY calculations in the Swedish setting.

5. Page 15, line 17. Can the authors further elaborate on the different findings of ICER 512,198?

-Response: We had elaborated some on this: see below:

An explanation for the huge ICER is that the study compared an intervention that was more expensive but had little incremental effect compared to the control.

6. The result regarding CEAC is shown for the WTP threshold of 15,905 per QALY at 0.5 probability of cost-effectiveness. It would be helpful to show in the Figure as well as in the main text (result section, and comments on the findings on discussion section accordingly) the probability of cost-effectiveness at the current WTP threshold used (UK NICE).

-Response: We fully understand this comment but doing this in the results section would give the wrong information as the Swedish system has no such threshold. The information presented is supposed to guide policy makers in their decisions (which every country has their own) and the UK threshold is only applicable in the UK so we would like not to change according to your comment and hope this is acceptable. This decision is grounded based on discussions with two different health economists in our department, whereof one is a co-author (Magnus Husberg). However, we chose to use the UK threshold in the discussion section as a comparator to give the reader something to refer to.

Minor

1. Page 5, line 18. Please include references to the study indicated in the sentence.

- Response: The reference is now provided.

2. Page 6, line 37. Revise the sentence to: In total, 144 patients fulfilled the eligibility criteria of individuals age>18 years and had no hospitalizations during the past four weeks prior to inclusions. Participants were (randomly?) allocated to nine weeks of iCBT (n=72) or ODF (n=72).

- Response: Thank your for these suggestions, that we now have applied.

3. Page 6, line 37. Add of: Comprised “of” goal setting...

- Response: This is added.

4. Page 7, line 41. Consider changing the sentence to: Six participants included in the primary RCT were residents in other regions that precluded costs data to be obtained. Hence, these patients are excluded from the analysis in this study.

- Response: Thank you for this suggestion, we have changed accordingly.

This sentence is also supposed to be placed in the result, not the method section.

-Response: We believe this sentence is better suited in the methods section and needed to inform the reader about the reason why the number of participants is lower than in the main RCT.

7. Page 11 line 13, place text (Figure 1) by end of the sentence.

- Response: This is done.

8. Page 14 line 20, change our to ours

- Response: We have rewritten this sentence to make it clearer that we were discussing our own study population, se below:

The CVD patients in the current study who had a mean age....

9. Page 14, line 19. Provide a reference for this sentence.

- Response: Please see previous response, the sentence is referring to our study and therefore a reference is not possible to provide.

10. Throughout, please make sure to place zero before point and make it consistent. E.g., change .05 to 0.05.

- Response: This is done.

11. Throughout, please carefully proofread for other grammatical errors, alignment and placing of references/ table/ figure by end of sentences.

- Response: This is done.

VERSION 2 – REVIEW

REVIEWER	Natsky, Andrea Flinders University, Health Economics, College of Medicine and Public Health
REVIEW RETURNED	19-Feb-2022
GENERAL COMMENTS	Thank you for addressing my queries. I found the authors' responses to be acceptable and well-reasoned. I recommend the study be published. Congratulations!